# Development of Commercial Eucalyptus Clone in Soil with Indaziflam Herbicide Residues

**Josiane Costa Maciel** [1,*], **Tayna Sousa Duque** [1], **Aline Cristina Carvalho** [2], **Brenda Thaís Barbalho Alencar** [2], **Evander Alves Ferreira** [3], **José Cola Zanuncio** [4], **Bárbara Monteiro de Castro e Castro** [4], **Francisca Daniele da Silva** [5], **Daniel Valadão Silva** [5] and **José Barbosa dos Santos** [1]

1. Departamento de Agronomia, Universidade Federal dos Vales do Jequitinhonha e Mucuri, Diamantina 39100-000, Brazil; taynaduque24@gmail.com (T.S.D.); jbarbosa@ufvjm.edu.br (J.B.d.S.)
2. Departamento de Engenharia Florestal, Universidade Federal dos Vales do Jequitinhonha e Mucuri, Diamantina 39100-000, Brazil; ninecarvalho87@gmail.com (A.C.C.); barbalhobrenda@gmail.com (B.T.B.A.)
3. Instituto de Ciências Agrárias, Universidade Federal de Minas Gerais, Montes Claros 39404-547, Brazil; evanderalves@gmail.com
4. Departamento de Entomologia/BIOAGRO, Universidade Federal de Viçosa, Viçosa 36570-900, Brazil; zanuncio@ufv.br (J.C.Z.); barbaramcastro@hotmail.com (B.M.d.C.e.C.)
5. Departamento de Manejo Solo e Água, Universidade Federal Rural do Semi-Árido, Mossoró 59625-900, Brazil; danieleamancio20@gmail.com (F.D.d.S.); daniel.valadao@ufersa.edu.br (D.V.S.)
* Correspondence: josi-agronomia@hotmail.com; Tel.: +55-38-99171-6384

**Abstract:** The pre-emergent herbicide indaziflam is efficient in the management of weeds in eucalyptus crops, but this plant may develop less in soil contaminated with it. The objective was to evaluate the levels of chlorophylls a and b, the apparent electron transport rate (ETR), growth and dry mass of leaves, stems and roots of Clone I144, in clayey soil, contaminated with the herbicide indaziflam and the leaching potential of this herbicide. The design was completely randomized in a 3 × 5 factorial scheme, with four replications. The leaching of indaziflam in the clayey soil profile (69% clay) was evaluated in a bioassay with *Sorghum bicolor*, a plant with high sensitivity to this herbicide. The injury and height of this plant were evaluated at 28 days after sowing (DAS). We believe that this is the first work on *Eucalyptus* in soil with residues of the herbicide indaziflam. Chlorophyll a and b contents and ETR, and height and stem dry mass of Clone I144, were lower in soil contaminated with indaziflam residues. The doses of indaziflam necessary to cause 50% ($C_{50}$) of injury and the lowest height of sorghum plants were 4.65 and 1.71 g ha$^{-1}$ and 0.40 and 0.27 g ha$^{-1}$ in clayey soil and sand, respectively. The sorption ratio (SR) of this herbicide was 10.65 in clayey soil. The herbicide indaziflam leached up to 30 cm depth at doses of 37.5 and 75 g ha$^{-1}$ and its residue in the soil reduced the levels of chlorophylls a and b, the apparent ETR and the growth of Clone I144.

**Keywords:** clonal eucalyptus; herbicide; indaziflam; leaching; soil profile

## 1. Introduction

The global demand for wood and wood products, a demand increasingly met by high-yield forest plantations, has been steadily growing [1]. These plantations have grown by an average of 4.4 million hectares annually, from 168 million hectares in 1990 to approximately 278 million hectares in 2015 [2]. To ensure high biomass production, significant quantities of agricultural materials are used [3]. Meeting these demands requires a dramatic increase in the global production and trade of forest products. This would imply a further increase in the global forest plantation area by about 25–67 million hectares to reach 303–345 million hectares by 2030, and there are predictions that the demand for roundwood supplied by forest plantations will increase by about 65% by 2070 [4].

*Eucalyptus* sp. is the most widely planted forest genus, with 25 million hectares [5,6] containing more than 110 species introduced in over 90 countries [7]. Brazil is the world leader

in eucalyptus planted area, followed by China and India [8,9]. It has 9.93 million hectares of planted forest, of which 75.8% is eucalyptus plantation [10]. Furthermore, Brazil is a leader in productivity, with an average accumulated mass of 40 $m^3ha^{-1}$ $year^{-1}$ [11], which has grown in recent years along tropical agricultural frontiers. Currently, the distribution and growth of eucalyptus plantation areas in Brazil is located in the Southeast, in Minas Gerais (30%) and São Paulo (13%), the Midwest, in Mato Grosso do Sul (14%), the Northeast, in Bahia (8%) and the South, in Rio Grande do Sul (8%) and Paraná (6%) [10]. Primary products, such as paper, pulp and wood, as well as secondary products, such as flooring and furniture, from Brazilian eucalyptus plantations are exported to many countries, highlighting the importance of Brazilian plantations for the international market [12].

Pure eucalyptus species, ranked in terms of importance [10], are mainly used in Brazilian plantations: *Eucalyptus grandis* (W. Hill ex Maiden), *Corymbia citriodora* (Hook.) KD Hill & LAS Johnson (formerly known as *E. citriodora*– basionym), *E. urophylla* (ST Blake), *E. saligna* (Sm.), *E. globulus* (Labill.), *E. camaldulensis* (Dehnh.), and hybrids *E. urophylla* × *E. grandis*, *E. urophylla* × *E. camaldulensis*, *E. grandis* × *E. camaldulensis* and *E. urophylla* × *E. globulus* [13,14]. These species or hybrids are selected for their characteristics, such as fast growth, wood quality, high productivity, profitability, strong adaptability to different soils and climatic conditions and ease of management [6,7]. We can also highlight an extensive history of investment in Brazil, and consolidated improvement techniques for silvicultural practices and forest genetic improvement.

Although the genetic improvement of this crop is at an advanced stage, another determining factor for the higher productivity of eucalyptus plantations is the control of diseases, pests and weeds [15,16]. Competition with weeds is a limiting factor for the development of most forest species [17]. Generally, weeds are considered the pest of greatest economic impact and phytosanitary risk in eucalyptus cultivation. Weeds seriously affect plant growth through interspecific competition for water, light and nutrients [18], causing serious damage to crop establishment, development and productivity. Although eucalyptus has potentially rapid growth rates, its tolerance to weed interference during establishment is low. Yield reduction due to weeds is greatest up to two years after eucalypt planting, when weed management in these crops is highly dependent on herbicides [19]. According to Silva et al. [20], specific plants can be controlled through the use of herbicides and their mechanisms of action.

Chemical control using herbicides is commonly employed for weed control. This weed control method in eucalyptus plantations is fast and efficient [21], with lower labor requirements and greater effectiveness. The development of a selective, broad-spectrum action herbicide, applied during the pre-emergence of weeds, would improve weed management for this crop and favor eucalyptus silviculture [22]. However, the number of herbicides used is reduced with few records for this crop [23] and most registered herbicides not being selective for eucalyptus [22].

Although there are some species that can be used as green manure to remove herbicides from the soil, as they have the ability to accumulate chemical compounds in tissues [24]. One of the areas with the greatest need for research development involves the use of chemical products for weed control in forest plantations, since application failures and herbicide drift can be harmful to the tree component and cause toxicity to plants, such that chemical controls must be used with caution. This situation is concerning given the low selectivity of herbicides to eucalyptus plantations, which can cause losses early on during tree development, leading to productivity losses [25]. The drift of glyphosate herbicide, non-selective to eucalyptus, can cause phytotoxicity, deformed apices, strongly developed necrosis along the leaf edges and marked leaf senescence [26]: nicosulfuron reduced stem diameter increment and fluazifop-p-butyl + fomesafen limited shoot dry mass accumulation [27].

The herbicide indaziflam N-[(1R,2S)-2,3-dihydro-2,6-dimethyl-1H-inden-1-yl]-6-[(1RS)-1-fluoroethyl]-1,3,5-triazine-2,4-diamine, an inhibitor of cellulose biosynthesis belonging to the alquilazinas group, is used during pre-emergence to manage weeds in coffee, citrus,

sugarcane, pine and eucalyptus crops in Brazil [23]. The structural formula of indaziflam is described in Figure 1. This herbicide is safe for grape [28] and olive [29] crops with low solubility in water (0.0028 kg m$^{-3}$ at 20 °C), o Koc < 1000 mL g$^{-1}$ organic carbon, pKa = 3.5, log Kow at pH 4, 7 or 9 = 2.8, prolonged residual activity in the soil and half-life (t$_{1/2}$) greater than 150 days [30]. These features reduce the environmental impact from indaziflam leaching into the soil and contaminating the groundwater [31]. However, soil mobility in eucalyptus plantations and the tolerance of this plant to indaziflam are poorly understood, increasing the need to evaluate its residual effects, especially in planting rows [32]. Thus, we hypothesized that indaziflam soil residues would reduce eucalyptus development.

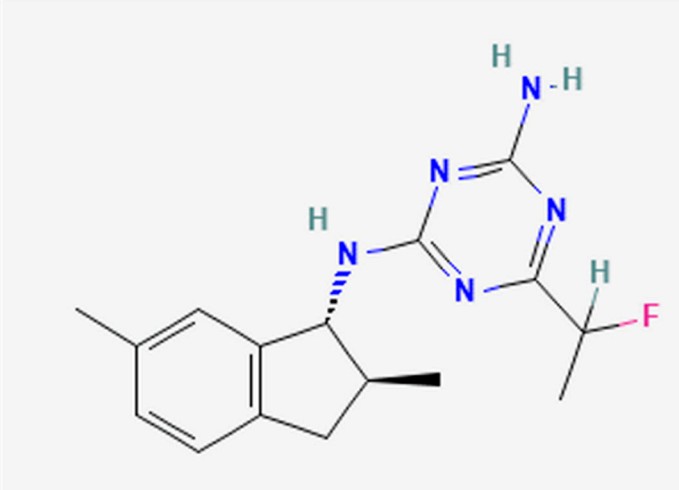

Source: PubChem (2023)

**Figure 1.** Representation of the chemical structure of indaziflam.

The objective was to evaluate the levels of chlorophyll a and b, ETR and the growth of Clone I144 in soil contaminated with indaziflam residues and the leaching potential of this herbicide.

## 2. Materials and Methods

The experiment was carried out in a greenhouse (minimum temperature of 25 °C and maximum temperature of 32 °C) at the Universidade Federal dos Vales do Jequitinhonha e Mucuri (UFVJM) in Diamantina, Minas Gerais, Brazil.

### 2.1. Experimental Design

The methodological design adopted in this study is outlined in Figure 2. The experiment had a completely randomized design, with treatments arranged in a 3 × 5 factorial scheme, with four replications. The first factor consisted of the control treatment (soil without herbicide) plus two doses, 35.7 and 75 g ha$^{-1}$, of Esplanade® herbicide (500 g a.i. L), with the doses corresponding to 25 and 50% of the commercially recommended dose for this product (150 g ha$^{-1}$). The second factor was the depth in soil profile: 0–10, 10–20, 20–30, 30–40 and 40–50 cm.

The eucalyptus clone used in the experiment was *Eucalyptus urograndis* (I144-*Eucalyptus urophylla* S.T. Blake × *Eucalyptus grandis* W. Hill ex Maiden). The eucalyptus clone was purchased in a nursery and was 45 days old. The clone was selected for its profitability, fast growth, high productivity and high-quality wood [15]. Each plot had a 150 mm PVC (polyvinyl chloride) tube, cut horizontally to form rings. The PVC columns were composed of five 10 cm high rings. Each one was filled with a sample of dystrophic red latosol (Table 1), previously fertilized as recommended for the crop.

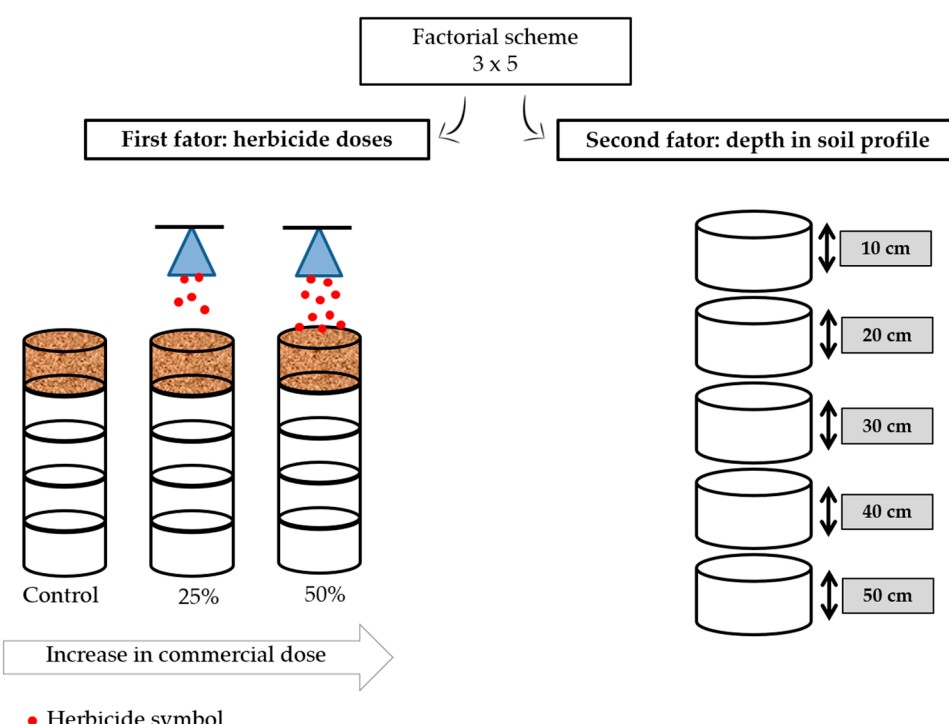

**Figure 2.** Diagram presenting the methodological design of the study.

**Table 1.** Physicochemical characteristics of the soil samples used in the experiment.

| Physical Analysis | | | | | | | | | | | | |
|---|---|---|---|---|---|---|---|---|---|---|---|---|
| Sand | | | Clay | | | Silt | | | Texture Class | | | |
| (dag kg$^{-1}$) | | | | | | | | | | | | |
| 6 | | | 69 | | | 25 | | | Very clayey | | | |
| Chemical analysis | | | | | | | | | | | | |
| pH | P | K | Ca | Mg$^{2+}$ | Al$^{3+}$ | H+Al | SB | t | T | V | m | OM |
| (H$_2$O) | (mg dm$^{-3}$) | | | | | (Cmol$_c$dm$^{-3}$) | | | | (%) | | (dag kg$^{-1}$) |
| 5.00 | 0.54 | 31 | 0.18 | 0.13 | 0.80 | 4.62 | 0.39 | 1.19 | 5.01 | 7.8 | 67.2 | 1.88 |

P-K-Extractor Mehlich 1; Ca-Mg-Al-Extractor: KCl-1 mol/L; H + Al-Calcium Acetate Extractor 0.5 mol/L-pH 7.0; SB = Sum of Bases; t = Effective Cation Exchange Capacity; T = Cation Exchange Capacity at pH 7.0; V = Base Saturation Index; m = Aluminum Saturation Index; OM = Organic Matter (C.Org × 1724-Walkley–Black).

### 2.2. Application of Indaziflam

Irrigation was carried out before the herbicide was applied, keeping humidity between 70% and 80% of field capacity. Indaziflam was applied with an electric sprayer (Yamaho FT5®, 5 L capacity) in a solution with a spray volume of 120 L ha$^{-1}$. The eucalyptus seedlings were transplanted one day after herbicide application, with one plant remaining per experimental unit. Irrigation was carried out using sprinklers, without exceeding the daily simulation limit of 60 mm of rain.

### 2.3. Chlorophyll Index and Electron Transport Rate

The chlorophyll index was determined using a chlorophyll meter (ChlorofiLOG CFL 1030®) between 9 a.m. and 10 a.m. on fully expanded leaves, at 14 days after planting, and the chlorophyll fluorescence was measured with a portable fluorometer ((MINI model) -PAM II, Walz, Effeltrich, Germany), at 21 days after planting, in expanded and photosynthetically active leaves, using specific leaf support tweezers (model 2030-B). This evaluation was performed at night with at least 30 min of adaptation of the leaves to the dark.

### 2.4. Height and Dry Mass of Leaves, Stems and Roots

The height of eucalyptus plants was measured with a ruler graduated in centimeters 120 days after planting. Leaves, stems and roots of this plant were conditioned in paper bags and dried in a forced air circulation oven (65 °C) for 48 h. The dry mass was determined on a precision scale.

### 2.5. Sorghum Bicolor as a Bioindicator Plant

*Sorghum bicolor* (L.) Moench hybrid BRS 655 (sorghum) was used as a bioindicator plant [32]. This sorghum species was planted in soil samples with known herbicide concentrations (dose–response curve). Indaziflam was applied at doses of 0, 0.25, 0.5, 1, 2, 3, 5, 10, 20, 40 and 60 g ha$^{-1}$, established in the sorghum sensitivity test to this herbicide [32], in dystrophic red latosol soil samples. Dose–response curves were plotted to evaluate sorghum cultivated in the soil. Ten sorghum seeds were sown, one day after herbicide application, in transparent plastic pots with a volume of 250 cm$^3$, an area of 50 cm$^2$, a height of 6 cm and a diameter of 10 cm. The thinning was performed after emergence, leaving six seedlings per pot. Pots under the same cultivation conditions were filled with soil samples from the eucalyptus experiment in order to estimate the residue by comparison with the dose–response curve. The pots were kept in a greenhouse under minimum temperature conditions of 15 °C, maximum of 35 °C and 75% humidity.

Sorghum plant injuries were visually assessed 28 days after sowing (DAS) using a scale from 0 to 100%, with 0% being no symptoms and 100% being plant death [33]. Plant height was measured in centimeters with a ruler. The indaziflam residue adsorbed into the soil was evaluated, simultaneously, in washed sand. The sand (0.6 mm to 2.0 mm) was washed in running water to remove impurities, immersed in an acid solution (10% sulfuric acid) for 24 h, and washed again in running water until the acid residue was removed. The pH was corrected to neutral (7) with the addition of sodium hydroxide solution (NaOH). The sand was dried in the sun on plastic sheeting for 24 h. The indaziflam doses estimated for the sand were 0, 0.05, 0.1, 0.15, 0.25, 0.5, 1, 2, 3, 5, and 10 g ha$^{-1}$ [32]. The sand volume and number of sorghum seeds were the same from the beginning to the end of the trial. The plants were irrigated with a nutrient solution (Table 2).

**Table 2.** Macro and micronutrients in the nutrient solution for irrigation of *Sorghum bicolor* in sand (CLARK 1975).

| Element | Source | Molecular Formula | Amount (mg L$^{-1}$) |
|---|---|---|---|
| N | Urea | $CH_4N_2O$ | 9.89 |
| P | Phosphoric acid | $H_3PO_4$ | 0.15 |
| K | Potassium chloride | KCl | 5.36 |
| Ca | Anhydrous calcium chloride | $CaCl_2$ | 11.56 |
| Mg | Magnesium chloride | $MgCl_2(6H_2O)$ | 4.82 |
| S | Sodium sulfate | $Na_2SO_4$ | 2.84 |
| B | Boric acid | $H_3BO_3$ | 0.05 |
| Cu | Copper chloride | CuCl | 0.003 |
| Fe | Iron chloride | $FeCl_3$ | 0.25 |
| Mn | Manganese chloride | $MnCl_2(4H_2O)$ | 0.056 |
| Zn | Zinc chloride | $ZnCl_2$ | 0.011 |
| Mo | Sodium molybdate | $Na_2MoO_4$ | 0.0052 |
| EDTA (5.44 g) + 0.824 g de NaOH | | | |

### 2.6. Statistical Analysis

All analyses were carried out using the R Core Team software version 3.4.3 with the R Studio software. Analysis of variance (ANOVA) using the F test and the Tukey test were used using the ExpDes.pt packages version 1.2.2 [34]. Regression analysis and 3D response surfaces were performed for injury and sorghum plant height. The significance

of the coefficients ($p < 0.05$) and the coefficient of determination were considered for the regression models. All statistical analyses were carried out at a 5% significance level.

The dose necessary to reduce the analyzed variable, injury or plant height, by 50% ($C_{50}$), was calculated for soil and sand, establishing a non-linear, log-logistic regression model with the equation of Seefeldt et al. [35]: $Y = C + D/1 + (X/C_{50})^{-b}$, where C = lower limit of the curve; D = difference between the upper and lower limits of the curve; b = slope of the curve; and $C_{50}$ = curve inflection point corresponding to 50% response. Graphs and $C_{50}$ were generated using SigmaPlot® (version 13.0, 2014, Systat Software, Inc., San Jose, CA, USA).

Indaziflam residue concentration by soil depth, was estimated by the percentage of visual injury of sorghum plants cultivated with soil depths of 0–10, 10–20, 20–30, 30–40 and 40–50 cm and with the $C_{50}$ of the analyzed variable.

The sorption ratio (RS) for indaziflam was calculated from the data obtained from soil $C_{50}$ in relation to sand, RS = $C_{50}$soil–$C_{50}$sand/$C_{50}$sand, which expresses the sorptive capacity of indaziflam into the soil, taking the soil and sand concentrations of the herbicide that inhibit 50% of the indicator plant's development as parameters.

## 3. Results

The results are outlined in Figure 3. Indaziflam reduced chlorophyll a and b levels, rate of electron transportation and height and dry mass of the eucalyptus plant stem. Indaziflam leached to a depth of 30 cm into clay soil (69% clay) at 121 days after application.

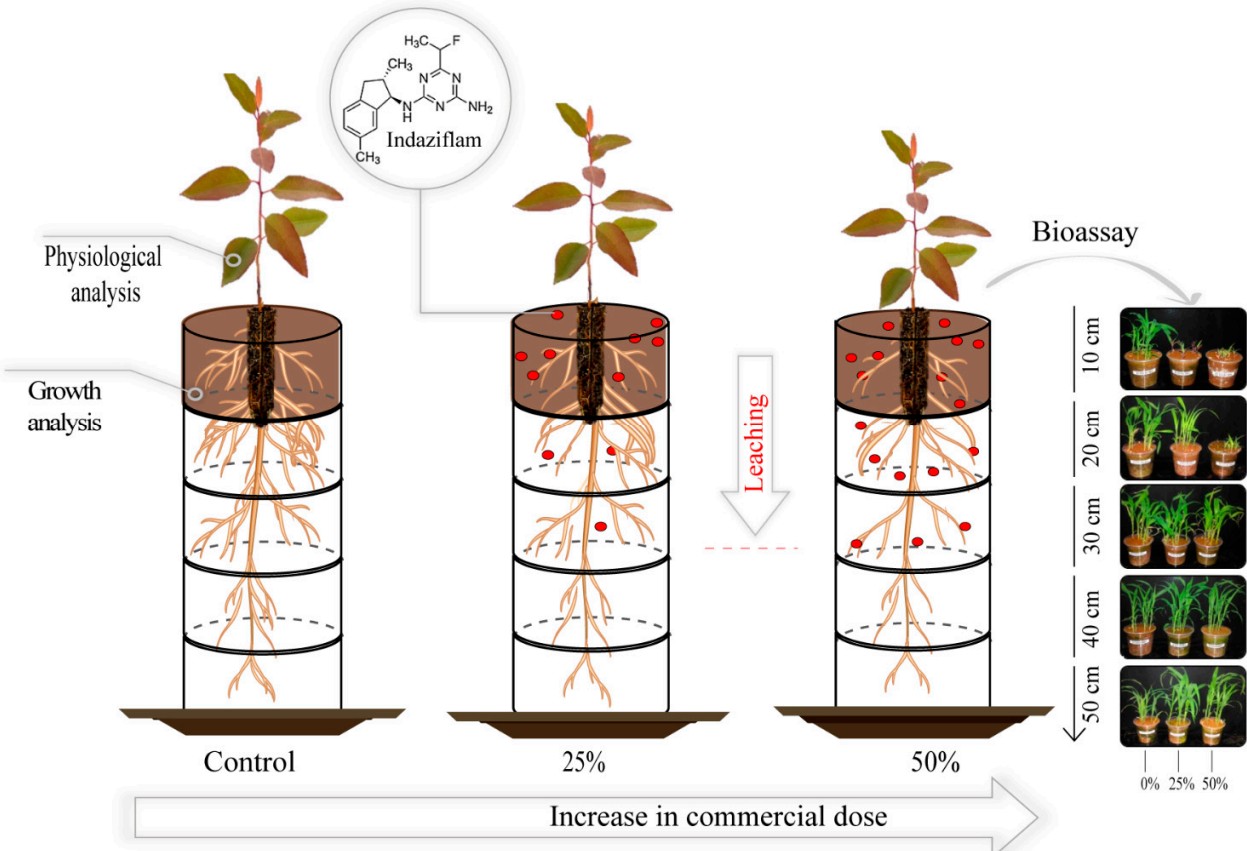

**Figure 3.** Representative scheme of the effect of indaziflam herbicide residues on the growth of Clone I144 (*Eucalyptus urophylla × Eucalyptus grandis*) at different soil depths.

### 3.1. Eucalyptus Plants

The chlorophyll a content of the Clone I144 was lower at 50% of the commercial indaziflam dose (Figure 4a), and the chlorophyll b content was lower at 25% and 50% of

the commercial indaziflam dose, than in the control, 14 days after planting (Figure 4b). The electron transport rate (ETR) of Clone I144 exposed to herbicide residues, was lower at 25% and 50% of the commercial dose 21 days after planting (Figure 4c).

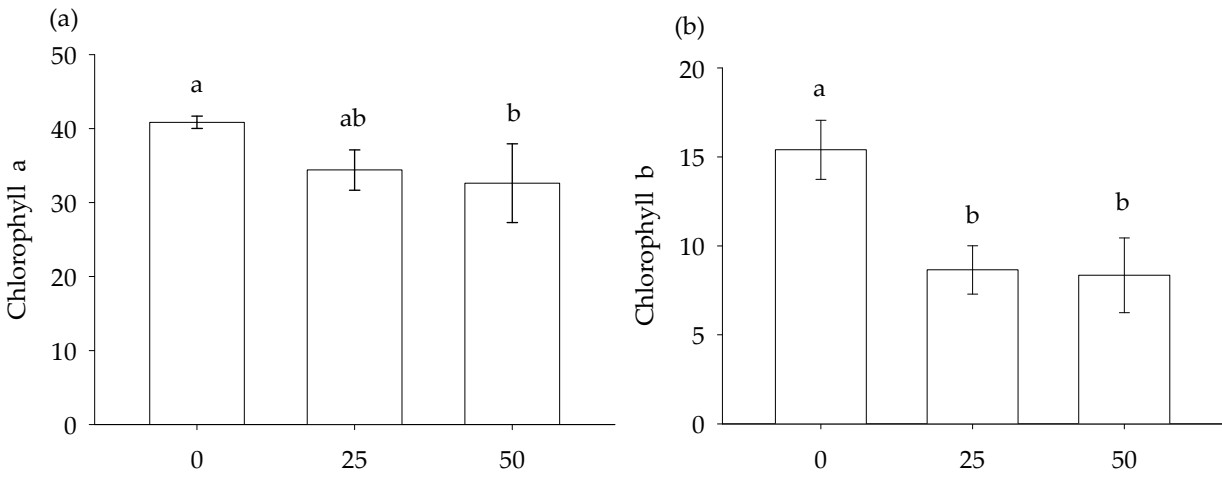

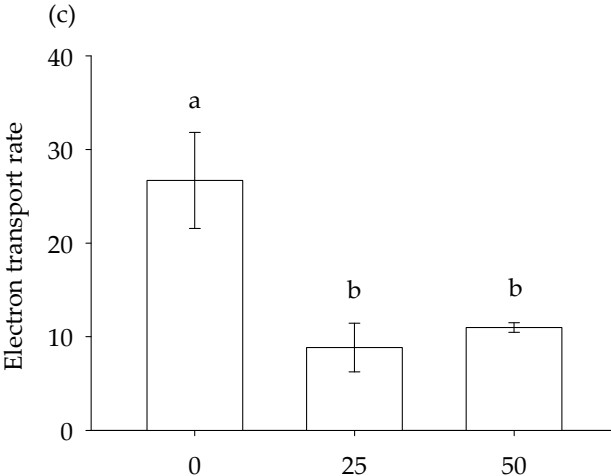

**Figure 4.** Chlorophyll (**a**,**b**) levels and electron transport rate (ETR) (**c**) of commercial eucalyptus clone I144 at 14 and 21 days after planting in soil contaminated with 25% and 50% of the commercial indaziflam dose (150 g ha$^{-1}$), respectively. Columns followed by the same lowercase letter, by parameter, do not differ by Tukey's test at 95% probability.

The height of Clone I144 was lower in soil contaminated with 25% and 50% of the commercial indaziflam dose, with a reduction of 12.46% under the effect of 50% of the commercial dose compared to the control (Figure 5).

Stem dry mass of Clone I144 was lower with 25% and 50% of the commercial indaziflam dose than in the control (Figure 6).

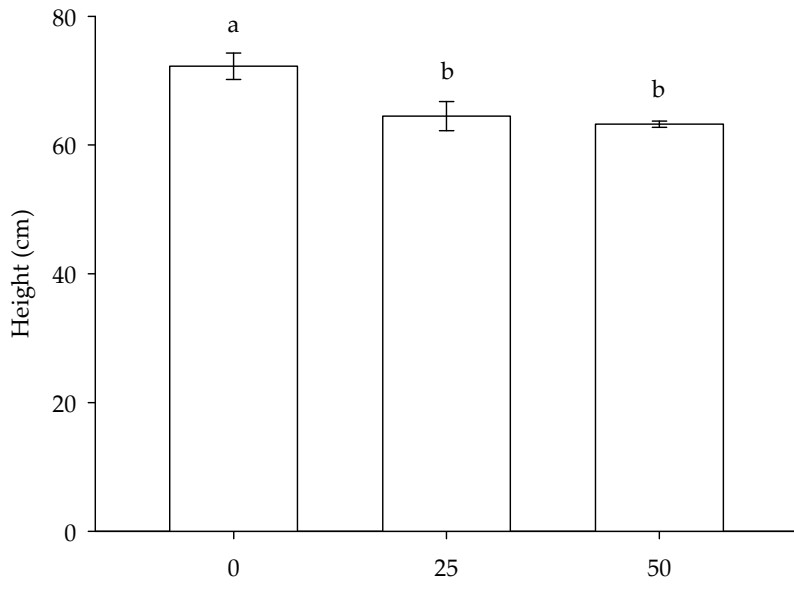

**Figure 5.** Height (cm) of commercial eucalyptus clone I144, 120 days after planting in soil contaminated with 25% and 50% of the commercial indaziflam dose (150 g ha$^{-1}$). Columns followed by the same lowercase letter do not differ according to Tukey's test at 95% probability.

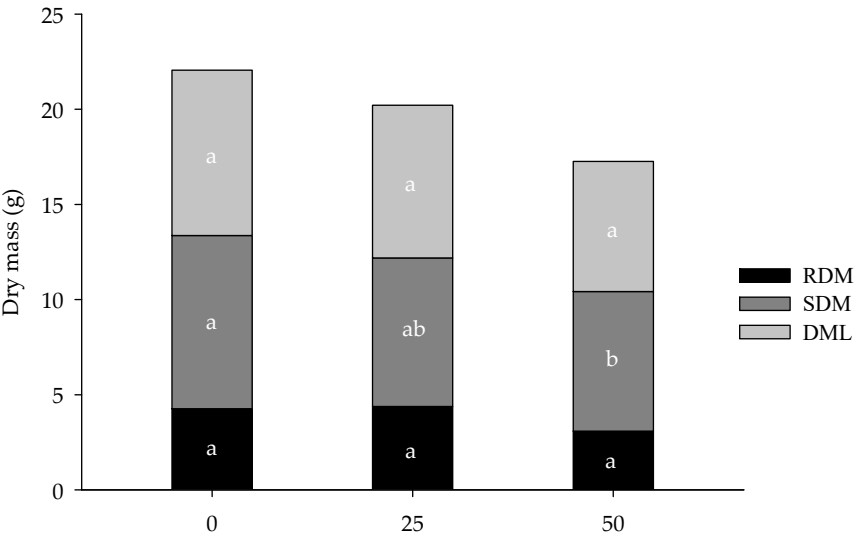

**Figure 6.** Dry mass of leaves (DML), stem (SDM) and roots (RDM) (g) of commercial eucalyptus clone I144, 120 days after planting in soil contaminated with 25% and 50% of the commercial indaziflam dose (150 g ha$^{-1}$). Columns followed by the same letter, per variable, do not differ by Tukey's test at 95% probability.

### 3.2. Sorghum Plants

The injury symptoms were maximal with soil removed at 15 cm and less than 10% with those at depths of 30–40 and 40–50 cm (Figure 7a). Sorghum plant height was lowest in soil contaminated with indaziflam up to 30 cm deep. Sorghum plant height variability was greater as a function of herbicide dose than of soil contamination depth, with the shortest height observed in soil contaminated up to 30 cm after being contaminated with the largest herbicide dose (Figure 7b). Initial symptoms observed in sorghum plants with increasing herbicide dose were leaf tissue reddening, leaf blade chlorosis and reduced growth.

(**a**) y = 89.427 + 0.605x − 5.327y + 0.003x² + 0.064y²     R² = 0.61     (**b**) y = −1.245 − 0.025x + 0.994y − 0.001x² − 0.011y²     R² = 0.68

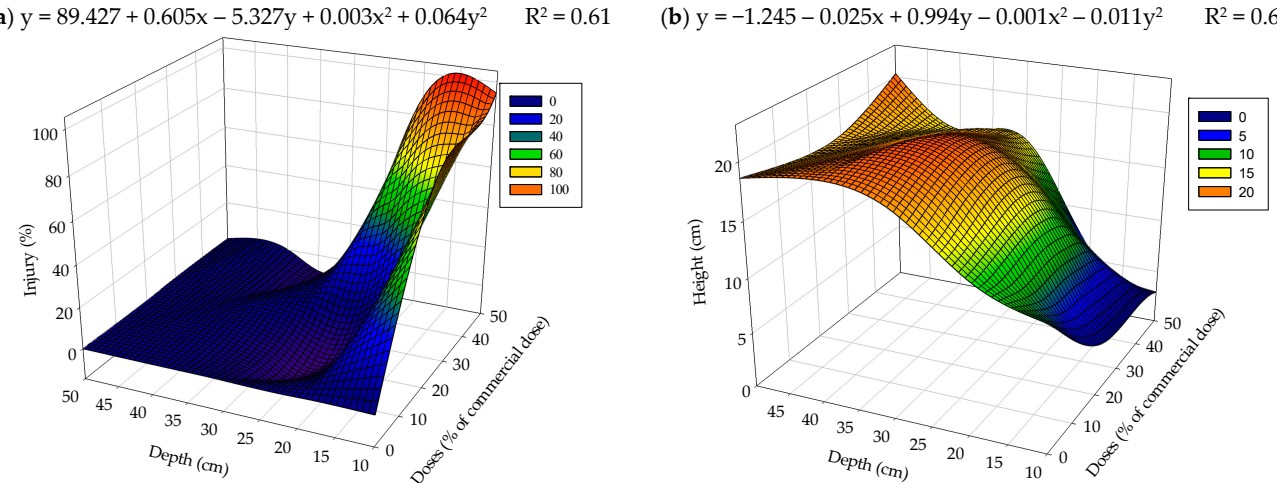

**Figure 7.** Injury (**a**) and height (**b**) of sorghum plants at 28 DAS as a function of indaziflam dose and soil depth cultivated with eucalyptus for 120 days.

The indaziflam doses necessary to cause 50% injury and reduced sorghum plant height, were 4.65 and 1.71 g ha$^{-1}$ in soil and 0.40 and 0.27 g ha$^{-1}$ in sand, respectively (Figure 8a,b).

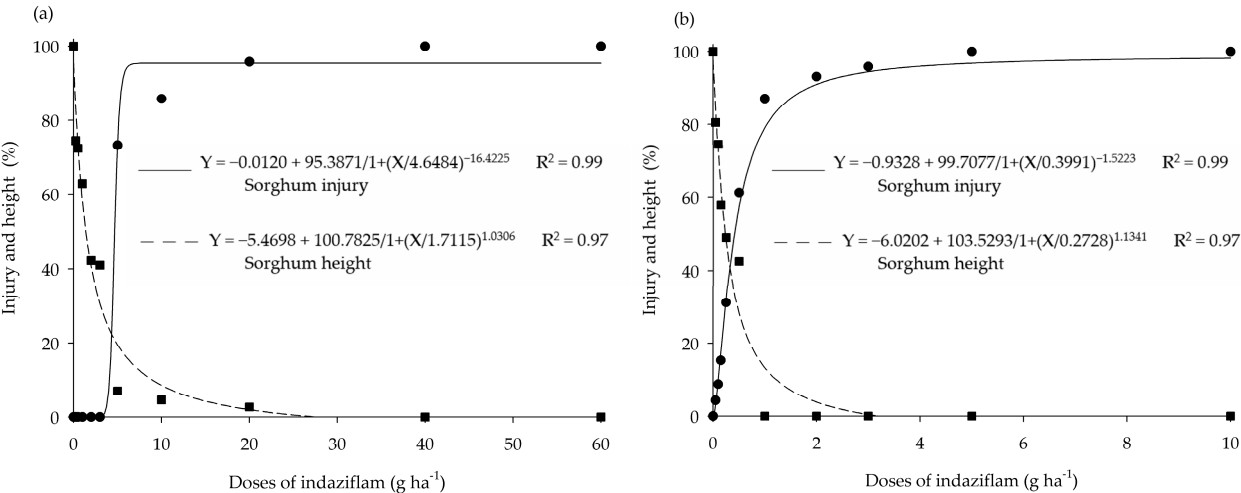

**Figure 8.** The dose–response curve for sorghum plant injury and height at 28 DAS grown in soil (**a**) and sand (**b**) with indaziflam doses of 0, 0.25, 0.5, 1, 2, 3, 5, 10, 20, 40 and 60 g ha$^{-1}$ and 0, 0.05, 0.1, 0.15, 0.25, 0.5, 1, 2, 3, 5, and 10 g ha$^{-1}$.

### 3.3. Indaziflam Soil Residues

The indaziflam soil residues, collected at 0–10, 10–20, 20–30, 30–40 and 40–50 cm depth, with 25% and 50% of the commercial herbicide dose, were 7.79 and 8.72 g ha$^{-1}$, 5.12 and 7.44 g ha$^{-1}$, 2.33 and 2.79 g ha$^{-1}$, 0 and 0 g ha$^{-1}$, and 0 and 0 g ha$^{-1}$, respectively (Figure 9).

The sorption ratio (SR) of the herbicide from the data obtained from soil $C_{50}$ in relation to sand was 10.65 in clayey soil.

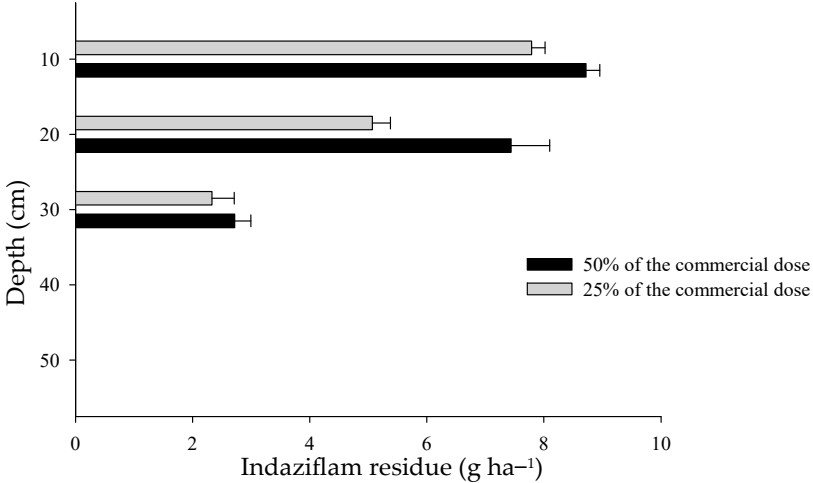

**Figure 9.** Indaziflam residue in soil samples cultivated with commercial eucalyptus clone I144 120 days after application of 25% and 50% of the recommended commercial dose (150 g ha$^{-1}$).

## 4. Discussion

Chemical control is a weed management alternative for forest plantations. Injuries caused by inadequate application, drift or herbicide soil residues are among the main problems reported when chemical control is employed [36]. Some herbicide use for weed control has a residual effect in the soil and can reduce the physiological and growth characteristics of the crop, affecting productivity.

The lower chlorophyll a and b levels at 50%, and 25 and 50%, respectively, of the commercial indaziflam dose can be explained by the indirect interference of the herbicide, affecting the translocation of divalent cations, such as magnesium (Mg) and manganese (Mn) [37], to the meristematic tissues, thereby inhibiting photosynthetic activity. Mg and Mn are essential for photosynthetic light reactions. Previous studies reported that the absorption bands of both chlorophylls a and b were directly related to the emission spectra of Mg [38]. The lower ETR of eucalyptus in soil contaminated with indaziflam is the result of the indirect effect of this herbicide on chlorophyll, reducing the emission of fluorescence signals by plant leaves [37]. ETR is a variable closely correlated with chlorophyll content [39]. Thus, lower chlorophyll levels reduce photon absorption and, consequently, lower ETR to the photosystem II binding site. Although the phytotoxic effects of indaziflam do not require light [40], it has been proposed that, as a cellulose biosynthesis inhibitor, it inhibits photosystem II [41].

The lower height of Clone I144 in soil contaminated with 25% and 50% of the commercial indaziflam dose is related to the action mechanism of this herbicide [32]. Indaziflam inhibits cellulose biosynthesis in plants [42,43], which is considered to be the main source of rigidity and structural support for plant cell walls [44,45]. Several accessory proteins are necessary for cellulose production and deposition, including Cellulose Synthase A (CESA), Korrigan, Cobra and Cellulose Synthase Interacting1 [46]. Loss of function in any of the necessary cellulose synthase subunit proteins causes complete or partial loss of anisotropic growth in expanding cells [40]. Interestingly, all these proteins are potential action sites for herbicides that inhibit cellulose biosynthesis [47]. These effects are seen not only in grasses [48], but symptoms have also been reported in perennial species such as macauba [49], sweet potato [43], *Coffea arabica* cultivar IBC12 [50] and pecan [51]. The lower height of Clone I144 at the highest dose (50% of the commercial dose) demonstrates the plant's susceptibility to lower than commercially recommended doses.

One potential explanation for the lower stem dry mass in the treatment with 50% of the commercial herbicide dose compared to the control is due to the action of the product, which inhibits cellulose biosynthesis, thereby leading to loss of integrity of the primary and secondary cell wall, formed by thin and thick layers of cellulose microfibrils [42]. It has already been reported that indaziflam inhibits the cellulose microfibril cross-linking

stage [50], reducing cell formation and consequently, plant dry mass. In trees, the secondary vascular tissues come from the activity of the secondary meristems that promote secondary growth in stem thickness. However, the effect of the herbicide on cellulose microfibril cross-linking and the inhibition of crystal deposition in the cell wall affect cell formation, division and elongation [52], leading to reduced stem dry mass. This may explain why the stem was the element most affected by indaziflam. The results of this study agree with previous research that shows varying lesions on the trunk of pecan plants between three and four months after indaziflam application [51].

Indaziflam leached presenting a residual effect up to a depth of 30 cm, and the symptoms observed in the bioindicator plant demonstrate the presence of the herbicide in numerous soil layers and their high sensitivity to indaziflam. The symptoms observed in sorghum plants, such as chlorosis of the young tissues, reddening of leaf tissue, necrosis and plant death, are characteristic of sensitive species exposed to herbicides that inhibit cellulose biosynthesis [32]. The injury and lower height of the sorghum plants at higher herbicide doses at a depth of up to 30 cm is due to direct herbicide action that inhibits cellulose biosynthesis [42], which can promote polymerization of cellulose from the UDP-glucose substrate by glucosyltransferase and also by inhibiting cell multiplication of other polysaccharides due to nitric acid accumulation [52]. Furthermore, cell division inhibition in meristematic tissue has also been suggested as a secondary mode of action that reduces cell formation and, consequently, plant height [53].

The value of the $C_{50}$ dose of 0.40 g ha$^{-1}$ for damage to sorghum plants grown in sand was also observed in a bioassay study [32]. This is the result of an inert substrate, in which the physical and chemical characteristics, such as the absence of organic matter, surface loads and clay, make it impossible for the herbicide to sorb, resulting in availability of the substrate and absorption by the plant roots [54]. This explains the greater injury and lower height of sorghum plants in sand than in soil.

The sorption ratio (SR) of 10.65 indicates a high amount of adsorbed indaziflam residue. Thus, the sorption ratio evaluated may be directly relate to the high clay content (69 dag kg$^{-1}$), organic matter (1.88 dag kg$^{-1}$) and pH (5.00) of the soil used in the study, which are similar to those reported in an experiment with Red-Yellow Latosol with pH (5.1) and Cambissolo with (SR) equal to 10 [32]. Physical and chemical soil characteristics generate different sorption capacities for herbicides, especially mineralogy and organic matter content, which are attributes that are directly involved in the sorption process of these products, as they have three-dimensional sites responsible for the sorption of ionic and non-ionic herbicides that form hydrogen bonds with the herbicides [55]. Herbicides applied pre-emergence and those derived from weak acids, such as indaziflam, are more adsorbed in the soil solution at low pH [56,57].

In this context, soils with a high sand content cause the herbicide to move downwards through the soil profile, due to the greater number of macropores as well as the low clay and organic matter levels [56]. Understanding the behavior and destination of this herbicide in the soil, as well as potential contamination risks stemming from variable soil properties, is important when explaining the possible presence of indaziflam in the planting lines affecting the crop.

## 5. Conclusions

Eucalyptus Clone I144 was sensitive to the herbicide indaziflam. The herbicide reduced chlorophyll a and b levels, the electron transport rate, and the height and dry mass of the stem of the clone evaluated. It leached to a depth of 30 cm at doses of 37.5 and 75 g ha$^{-1}$. This is the first report of the effects of indaziflam residue on the physiological and growth characteristics of a eucalyptus clone.



**Author Contributions:** J.C.M.: conceptualization, formal analysis, investigation, writing—original draft, writing—review and editing. T.S.D.: formal analysis, investigation, writing—original draft, writing—review and editing. A.C.C.: formal analysis, investigation, writing—original draft. B.T.B.A.: formal analysis, investigation, writing—original draft. E.A.F.: conceptualization, methodology, resources, writing—review and editing. J.C.Z.: resources, writing—original draft, writing—review and editing, supervision. B.M.d.C.e.C.: resources, writing—original draft, writing—review and editing, supervision. F.D.d.S.: writing—review and editing. D.V.S.: writing—review and editing, resources. J.B.d.s.: conceptualization, methodology, resources, writing—original draft, writing—review and editing. All authors have read and agreed to the published version of the manuscript.

**Funding:** This research was funded by Conselho Nacional de Desenvolvimento Científico e Tecnológico (CNPq), Programa MAI DAI UFVJM, Coordenação de Aperfeiçoamento de Pessoal de Nível Superior (CAPES)—Código Financeiro 001 and Fundação de Amparo à Pesquisa do Estado de Minas Gerais (FAPEMIG).

**Data Availability Statement:** Data is contained within the article.

**Acknowledgments:** To the "Conselho Nacional de Desenvolvimento Científico e Tecnológico (CNPq)", "Coordenação de Aperfeiçoamento de Pessoal de Nível Superior (CAPES)—Código Financeiro 001" and "Fundação de Amparo à Pesquisa do Estado de Minas Gerais (FAPEMIG)" and "Programa Cooperativo sobre Proteção Florestal (PROTEF) do Instituto de Pesquisas e Estudos Florestais (IPEF)" for financial support. Phillip John Villani (University of Melbourne, Australia), a professional editor and proofreader and native English speaker, has reviewed and edited this article for structure, grammar, punctuation, spelling, word choice, and readability.

**Conflicts of Interest:** The authors declare that they have no conflict of interest.

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
