# Peer review of "Development of Commercial Eucalyptus Clone in Soil with Indaziflam Herbicide Residues"

_forests, doi:10.3390/f14091923_

Round 1

Reviewer 1 Report

The manuscript presents the effect of the herbicide indaziflam on a eucalyptus biotype and herbicide leaching potential. The work needs mandatory corrections. Some general comments:

Introduction = Sentences in lines 49 and 50 were not in English.

Material and methods = The experimental design was confused. Figure 1 helps to understand and could be used in this section. There is confusion in the units of chlorophyll (the correct one is chlorophyll index). The term “intoxication” is used erroneously, the correct word would be “injury” for visual damage caused by herbicides. It was not clear how the quantification of the indaziflam residue was carried out. Additional information such as the amount of water added during the experiment is important to analyze the leaching potential.

Results = Figure 3 needs to be improved, information is missing and it seems to be out of standard in relation to Figures 2 and 4.

Discussion = The discussion of the results was very poor. There is confusion about the direct effect of the herbicide indaziflam with indirect effects. Data from “levels of chlorophylls” and “ETR” can be discussed in sequence, as they are complementary. Likewise the growth data of Clone I144. In the sorghum section, it was necessary to deepen the indaziflam residual data, exploring the literature on soil types, leaching, etc.

Other comments are in the file.

The quality of English needs to be improved. There are parts of the manuscript in Portuguese.

Author Response

Dear Dr. Reviewer,

Follow the manuscript “forests-2529520” for your appreciation. In this new version, all the suggestions made by the reviewers were made and the questions answered. Below are all the changes made to the point-to-point manuscript, they were highlighted in red in the text.

Please don't hesitate to contact me for further information. Thanks in advance for your time and consideration.

Kind regards,

Josiane Costa Maciel*

*Corresponding author: josi-agronomia@hotmail.com

Reviewers' comments:

Reviewer #1:

Reviewer’s comment: The manuscript presents the effect of the herbicide indaziflam on a eucalyptus biotype and herbicide leaching potential. The work needs mandatory corrections. Some general comments:

Introduction = Sentences in lines 49 and 50 were not in English.

Reply: First of all, we must express our appreciation to you for the detailed feedback on our manuscript. We believe that the feedback has resulted in a substantially improved paper. We carefully analyzed each of the points made by the reviewer and tried to carefully answer each one of the recommendations.

We thank the reviewer for this important observation. This correction has been made in the text (lines 48-49).

Reviewer’s comment: Material and methods = The experimental design was confused. Figure 1 helps to understand and could be used in this section. There is confusion in the units of chlorophyll (the correct one is chlorophyll index). The term “intoxication” is used erroneously, the correct word would be “injury” for visual damage caused by herbicides. It was not clear how the quantification of the indaziflam residue was carried out. Additional information such as the amount of water added during the experiment is important to analyze the leaching potential.

Reply: In response to “The experimental design was confused. Figure 1 helps to understand and could be used in this section”. Done. Thank you for this suggestion. We modified the paragraph to make it clearer. We corrected the text (lines 123-129). We have also added a methodological design, Figure 2.

In response to “There is confusion in the units of chlorophyll (the correct one is chlorophyll index)”. Done. These corrections were made in the text.

In response to “The term “intoxication” is used erroneously, the correct word would be “injury” for visual damage caused by herbicides”. Done. We agree with your assertation. We have removed the word "intoxication". We have added the word "injury" to the text.

In response to “It was not clear how the quantification of the indaziflam residue was carried out”. The herbicide indaziflam was quantified by assessing damage to sorghum plants grown in soil samples from each treatment. To estimate the residue, the damage was compared within a scale of effects in a dose response curve constructed with 10 known concentrations of the herbicide (0, 0.25; 0.5; 1; 2; 3; 5; 10; 20; 40 and 60 g ha-1). This information was added to the text (lines 167 to 168).

In response to “Additional information such as the amount of water added during the experiment is important to analyze the leaching potential”. Thank you for this suggestion. We've reworded the paragraph to make it broader and more detailed. We've added this information on lines 147-152.

Reviewer’s comment: Results = Figure 3 needs to be improved, information is missing and it seems to be out of standard in relation to Figures 2 and 4.

Reply: Done. Thank you for this suggestion. We removed figure 2 and made a new one Figure (Figure 5).

Reviewer’s comment: Discussion = The discussion of the results was very poor. There is confusion about the direct effect of the herbicide indaziflam with indirect effects. Data from “levels of chlorophylls” and “ETR” can be discussed in sequence, as they are complementary. Likewise the growth data of Clone I144. In the sorghum section, it was necessary to deepen the indaziflam residual data, exploring the literature on soil types, leaching, etc.

Reply: Done. Thank you for this suggestion. We made all the corrections and rewrote the discussion.

Reviewer’s comment: Other comments are in the file.

Reply: Done. Thanks for the comment. All the comments in the file have been corrected and they were highlighted in red in the text.

In response to the file comment “Please, translate to English”. Done. This correction has been made in the text.

In response to the file comment “Confused. Change to "two doses 35.7 and 75 g ha-1 of Esplanade(R) herbicide (500 g a.i. L)"”. Done. Thank you for this suggestion. This correction has been made in the text.

In response to the file comment “Was the herbicide applied before or after transplanted??”. The herbicide was applied before the seedlings were transplanted in order to simulate a soil with indaziflam herbicide residue.

In response to the file comment “Did the irrigation start before or after the application of indaziflam?”. The soil was irrigated before the herbicide was applied.

In response to the file comment “Why were the analyzes done in different periods??”.  All the analyses were carried out during the same period. The variables were evaluated at 14 e 21 days after planting, but we preferred to keep only the results with statistical differences.

In response to the file comment “Has the "injury" effect of indaziflam on eucalyptus plants been analyzed?”. No. It has not been analyzed.

In response to the file comment “dry weight or dry matter??”. This correction has been made in the text. We've removed the word "dry weight" and added "dry mass".

In response to the file comment “Clarify, is the soil used in the sorghum experiment the same as that used in the previous experiment?”. Yes. The soil used in the sorghum experiment was a dystrophic red latosol, the same soil used in the previous experiment. These corrections were made in the text.

In response to the file comment “The term "intoxication" is not recommended for assessing the effects of herbicides. Change to "injury" or "crop injury"”. Done. Thank you for this suggestion. We've removed the word "intoxication" and added "injury".

In response to the file comment “The regression models were performed on which package?”. We modified the paragraph to make it clearer. We corrected the text. Please also see lines 192-197.

In response to the file comment “That's a discussion, I suggest moving on to the next section”. Done. Thank you for this suggestion. We've added the phrase to the Discussion section.

In response to the file comment “How long after application?”. “How much water was added?”. It was 121 days after the application. Irrigation was carried out by sprinklers, without exceeding the daily simulation limit of 60 mm of rain. We have added this information in the text.

In response to the file comment “The figure should be in the material and methods, as it helps in understanding how the experiment was set up”. In the Material and Methods section, we have added a figure showing the methodological design of the study. See also Figure 2. Figure 3 is intended to represent the Results.

In response to the file comment “Please check the units. For Material and methods, a chlorophyll meter was used, therefore the reading is given in the "chlorophyll index"”. Thanks for the comment. We have removed this information from the figure legend.

In response to the file comment “Please add the error bar in both directions (top and bottom)”. Done. Thank you for this suggestion. We've added the error bar in both directions. See also figures 4 and 5.

In response to the file comment “Confusing and uninformative. Is the line a linear regression? If so, where are the equation parameters? I suggest keeping the pattern of Figures 2 and 4 and presenting it in column form with error bars”. Done. Thank you for this suggestion. Let's redo the figure. See Figure 5.

In response to the file comment “Change the Y-axis title, default to just "Dry matter (g)”. Done. We removed the word “Dry matter”. We added the word “Dry mass”.

In response to the file comment “The sentence is misplaced. I suggest leaving it in the sequence after the injury assessment”. Done. Thank you for this suggestion. Please also see lines 251-252.

In response to the file comment “The abbreviation has already been described in the material and methods, just use the abbreviation”. Done. Thank you for this suggestion.

In response to the file comment “It was not clear in the methodology how the quantification of the indaziflam residue was performed. Please clarify”. Done. We have added this information in the text.

In response to the file comment “This is not correct. The herbicide does not act directly on this route. The observed effects may be due to an indirect effect of indaziflam action. Please clarify”. Done. We reformulated the Discussion to be broader and cite what's new. We've reworded the paragraph.

In response to the file comment “I disagree with the sentence. This may be an indirect effect of indaziflam, however, inhibition of cellulose biosynthesis and consequently inhibition of growth are the most likely direct effects. Please rewrite the paragraph”. Done. We've reworded the paragraph.

Reviewer’s comment: The quality of English needs to be improved. There are parts of the manuscript in Portuguese.

Reply: We thank the reviewer for this important observation. The manuscript was edited for proper English language, grammar, punctuation, spelling, and overall style by one or more of the highly qualified native-speaking English.

We appreciate this reviewer's suggestions.

Reviewer 2 Report

The manuscript entitled “Development of commercial eucalyptus clone in soil with indaziflam herbicide residues” is an interesting study. Nevertheless, the manuscript's presentation and structure are inappropriate. It is necessary to make some important revisions. Prior to publication, the following revisions need to be considered:

General comments:

- There was a repetition of this sentence in abstract: “Chlorophyll a and b contents, apparent electron transport rate (ETR), height growth and dry matter of leaves, stems and roots of Clone I144 were evaluated.”. Rewrite the abstract.

- Introduction lines 49-50: According to the context, this sentence is in Portuguese: “Eucalipto sp. é o gênero florestal mais plantado com 25 milhões de hectares [4, 5], contendo mais de 110 espécies introduzidas em mais de 90 países”. It should be translated into English.

- Material and methods, lines 118-124: The presentation of this section is poor and the sentences are vague. It needs to be rewritten to make it more clear.

- Material and methods, statistical analysis: Which package of R did you use?

- Results, lines 199-202: The presented text is not a result and should be removed from this section.

- Figure 1: Present more quality version for this figure.

- The discussion section is very poor and not acceptable as required for a research paper.

- Conclusions: Just repeating the results! Rewrite it.

There is a major problem with the English language quality. Some of the sentences are vague and should be taken into consideration. There were some sentences presented in Portuguese by the authors (lines 49-50).

Author Response

Dear Dr. Reviewer,

Follow the manuscript “forests-2529520” for your appreciation. In this new version, all the suggestions made by the reviewers were made and the questions answered. Below are all the changes made to the point-to-point manuscript, they were highlighted in red in the text.

Please don't hesitate to contact me for further information. Thanks in advance for your time and consideration.

Kind regards,

Josiane Costa Maciel*

*Corresponding author: josi-agronomia@hotmail.com

Reviewers' comments:

Reviewer #2:

Reviewer’s comment: The manuscript entitled “Development of commercial eucalyptus clone in soil with indaziflam herbicide residues” is an interesting study. Nevertheless, the manuscript's presentation and structure are inappropriate. It is necessary to make some important revisions. Prior to publication, the following revisions need to be considered:

General comments:

There was a repetition of this sentence in abstract: “Chlorophyll a and b contents, apparent electron transport rate (ETR), height growth and dry matter of leaves, stems and roots of Clone I144 were evaluated.”. Rewrite the abstract.

Reply: First of all, we must express our appreciation to you for the detailed feedback on our manuscript. We believe that the feedback has resulted in a substantially improved paper. We have undertaken all the changes.

We thank the reviewer for this important observation. We deleted the repeated phrase in the abstract. The Abstract has been rewritten.

Reviewer’s comment: Introduction lines 49-50: According to the context, this sentence is in Portuguese: “Eucalipto sp. é o gênero florestal mais plantado com 25 milhões de hectares [4, 5], contendo mais de 110 espécies introduzidas em mais de 90 países”. It should be translated into English.

Reply: Done. Thank you for detailed attention. We corrected this point in the Introduction. The sentence has been translated into English, please see lines 48-49.

Reviewer’s comment: Material and methods, lines 118-124: The presentation of this section is poor and the sentences are vague. It needs to be rewritten to make it more clear.

Reply: Done. Thank you for this suggestion. We agree with your assertation. We modified the paragraph to make it clearer. We corrected the text (lines 123-129). We have also added a methodological design, Figure 2.

Reviewer’s comment: Material and methods, statistical analysis: Which package of R did you use?

Reply: The ExpDes.pt package version 1.2.2 was used. We have added this information in the text (lines 192-197).

Reviewer’s comment: Results, lines 199-202: The presented text is not a result and should be removed from this section.

Reply: Done. Thank you for this suggestion. We've added the phrase to the Discussion section.

Reviewer’s comment: Figure 1: Present more quality version for this figure.

Reply: Done. Thank you for this suggestion. We delete the figure and add a new figure with better quality. See also Figure 3.

Reviewer’s comment: The discussion section is very poor and not acceptable as required for a research paper.

Reply: Done. Thank you for this suggestion. We made all the corrections and rewrote the discussion.

Reviewer’s comment: Conclusions: Just repeating the results! Rewrite it.

Reply: Done. Thank you for this suggestion. We have rewritten the Conclusion. See lines 361-365.

Reviewer’s comment: There is a major problem with the English language quality. Some of the sentences are vague and should be taken into consideration. There were some sentences presented in Portuguese by the authors (lines 49-50).

Reply: We thank the reviewer for this important observation. The manuscript was edited for proper English language, grammar, punctuation, spelling, and overall style by one or more of the highly qualified native-speaking English.

We appreciate this reviewer's suggestions.

Round 2

Reviewer 2 Report

Thanks for considering all my comments and suggestions.

You may just add explanation to Supplementary Materials section.